# The Role of Visual Information Quantity in Fine Motor Performance

**DOI:** 10.3390/jfmk9040267

**Published:** 2024-12-11

**Authors:** Giulia Panconi, Vincenzo Sorgente, Sara Guarducci, Riccardo Bravi, Diego Minciacchi

**Affiliations:** 1Department of Experimental and Clinical Medicine, University of Florence, 50134 Firenze, Italy; vincenzo.sorgente@unifi.it (V.S.); riccardo.bravi@unifi.it (R.B.); 2Department of Information Engineering, University of Florence, 50134 Firenze, Italy; sara.guarducci@unifi.it

**Keywords:** fine motor movements, fine motor behavior, visual feedback, sensorimotor integration, tracing task, graphic pen tablet, motor performance

## Abstract

Background/Objectives: Fine motor movements are essential for daily activities, such as handwriting, and rely heavily on visual information to enhance motor complexity and minimize errors. Tracing tasks provide an ecological method for studying these movements and investigating sensorimotor processes. To date, our understanding of the influence of different quantities of visual information on fine motor control remains incomplete. Our study examined how variations in the amount of visual feedback affect motor performance during handwriting tasks using a graphic pen tablet projecting on a monitor. Methods: Thirty-seven right-handed young adults (20 to 35 years) performed dot-to-dot triangle tracing tasks under nine experimental conditions with varying quantities of visual cues. The conditions and triangle shape rotations were randomized to avoid motor training or learning effects. Motor performance metrics, including absolute error, time of execution, speed, smoothness, and pressure, were analyzed. Results: As visual information increased, absolute error (from 6.64 mm to 2.82 mm), speed (from 99.28 mm/s to 57.19 mm/s), and smoothness (from 4.17 mm^2^/s^6^ to 0.80 mm^2^/s^6^) decreased, while time of execution increased (from 12.68 s to 20.85 s), reflecting a trade-off between accuracy and speed. Pressure remained constant across conditions (from 70.35 a.u. to 74.39). Spearman correlation analysis demonstrated a moderate to strong correlation between absolute error and time of execution across conditions. The Friedman test showed significant effects of experimental conditions on all motor performance metrics except for pressure, with Kendall’s W values indicating a moderate to strong effect size. Conclusion: These findings deepen our understanding of sensorimotor integration processes and could potentially have implications for optimizing motor skills acquisition and training and developing effective rehabilitation strategies.

## 1. Introduction

Fine motor movements encompass precise, goal-directed motor behaviors executed by small muscle groups, which require accurate and complex eye–hand coordination [1,2,3]. These movements are essential for a wide range of daily activities and professional tasks, from handwriting and tool manipulation to surgical procedures [4,5,6]. Understanding the factors influencing fine motor performance is therefore crucial for optimizing skill acquisition and rehabilitation strategies [5,7,8,9]. Furthermore, it is well established that motor and cognitive development are inextricably intertwined during infancy and early childhood [10,11,12]. For instance, fine motor impairments are very common in children with ADHD, and fine motor training programs of varying intensities have proven to be effective in improving motor skills, enhancing task performance, and potentially aiding overall cognitive and behavioral development [13]. Additionally, fine motor training can enhance neural activity in brain regions associated with the non-dominant hand in healthy adults [14].

Humans instinctively leverage all available sensory inputs to achieve precise motor performance [15]. Visual feedback, in particular, was shown to reduce motor errors by providing detailed information that facilitates refined motor responses and the increase in motor and neural complexity [6,16,17,18,19,20,21,22]. Indeed, as highlighted by previous studies, visual information plays a crucial role in planning, guiding, and refining movements [18,19,21,23,24,25,26,27]. Additionally, experiments involving targets of varying colors, shapes, and sizes have demonstrated the substantial impact of visual information quality on motor performance [7,28,29,30,31].

To further explore the critical role of visual feedback in fine motor performance, employing effective methodologies is essential. Amongst these, tracing tasks are commonly employed to provide valuable insights into the visuomotor system [29,30,32,33,34]. Tracing tasks are characterized by external visual guidance and allow us to delve into the concept of sensorimotor integration by adopting an ecological methodology [29,35]. Sensorimotor integration refers to the process by which sensory inputs, particularly visual feedback, are processed and used to guide and refine motor actions. This is an essential aspect of motor control, as it allows the brain to adapt movements based on the sensory information available. It has been hypothesized that variations in visual information has led to shifts in cueing strategies, where increased visual input prompts reliance on external cues (tracing), while reduced visual feedback necessitates greater dependence on internal cues (drawing) [29]. Differences in brain activity observed under varying quantities of visual information conditions suggest the significant role of visual information quantity in shaping motor performances during tracing tasks [32,33].

Several tracing tasks involving different shapes were previously used in the literature [29,33,34,36]. Among the various tracing tasks used in the literature, we selected the triangle tracing task for several key reasons aligned with the aim of our study. First, the triangle represents the simplest 2D geometric form, offering a controlled and simplified method for studying the influence of varying levels of visual feedback on motor performance. Its simplicity allows for more precise measurements of movement accuracy and control, without the confounding factors present in more complex, real-world tasks like handwriting. Second, triangle tracing tasks are highly reproducible and suitable for experimental manipulation, such as varying the amount of visual input and shape rotation, both of which are central to our study. Unlike handwriting, which can be influenced by individual writing styles and cognitive factors like language or memory, triangle tracing offers more consistent motor performance across participants. This consistency is crucial for isolating the effects of visual feedback on motor behavior. Additionally, previous studies have successfully used triangle tracing tasks to explore fine motor performance, making it a well-established model for investigating the effects of visual feedback on motor control [32,33,34].

Furthermore, various tools have been employed to study performance in these tasks, including accelerometers, electromyography, and force sensors [6,37]. However, despite our understanding of the influence of visual feedback on motor performance [11], an actual gap remains in our comprehension of how different quantities of visual information impact motor control, particularly in fine motor movements.

Our study aims to understand how variations in the amount of visual feedback impacts motor performance during dot-to-dot triangle tracing tasks using a graphic pen tablet projecting input on a monitor. This is achieved by studying motor performance parameters that are well established in the literature, such as absolute error and time of execution along with less commonly employed metrics, such as smoothness and pressure, with the purpose of providing a more comprehensive understanding of fine motor performance.

We hypothesized that variations in the amount of visual feedback during a tracing task will lead to significant differences in motor performance metrics, including execution time, accuracy, smoothness, and pressure. More specifically, we predicted that increased visual input will reduce errors and improve task efficiency, while reduced feedback will challenge the motor system, potentially altering the internal cueing mechanisms involved in sensorimotor integration.

By enlightening the relationship between the quantity of visual information and fine motor performance, we aim to provide novel insights into how sensory information processing influences motor control. Understanding this connection is crucial as it provides fundamental insights into the behavioral mechanisms and allows us to speculate on the neural processes underlying motor skill execution. Our findings are expected to contribute to the advancement of motor control research and its clinical applications through the optimization of motor skill acquisition and development of effective rehabilitation strategies.

## 2. Methods

### 2.1. Participants

This study employed a within-subject experimental design to investigate the impact of varying visual information on fine motor performance during a dot-to-dot tracing task.

To ensure an adequate sample size, a power analysis was conducted using G*Power 3.1 software. This analysis indicated that a minimum of 22 participants was necessary to detect a small effect size (Cohen’s f = 0.2 for ANOVA-type tests) with 0.80 power and a 0.05 significance level, justifying the inclusion of 37 participants in this study (0.97 power). We adopted a conservative approach by assuming a small effect size to ensure an adequate sample size for detecting meaningful differences across conditions. Therefore, we accepted 37 right-handed university students (20 males, 17 females, age range 20–35 years) who volunteered to participate. All participants were naive to the task, the purpose of the study, and the graphic pen setup used for the test. Participants reported no documented motor or neurological impairments. Moreover, participants attested normal or corrected-to-normal visual acuity and no previous experience with a graphic pen tablet. The study protocol was approved by the Institutional Ethics Committee (Area Vasta Centro AOU Careggi, Florence, Italy—ref:17768_oss). Before the start of the experiments, participants provided written informed consent.

### 2.2. Experimental Setup and Design

The task was executed using a graphic pen tablet (Wacom Intuos^®^ CTH-690AK, Tokyo, Japan, frequency rate: 133 Hz, resolution: 2540 lpi, active area: 216 × 135 mm, pressure levels 1024) and a monitor (HP, Palo Alto, United States, 27f Display, 1080 p, 68.6 cm, 27-inch Diagonal IPS, 75 Hz refresh rate) (Figure 1). The choice to use a pen tablet for the tracing task stems from its significant capacity to provide both ecological validity and precision [2,38]. Unlike traditional paper-and-pencil methods employed in clinical settings, a graphic pen tablet facilitates digital data collection, thereby allowing for more accurate and comprehensive measurements of motor performance [39,40]. Also, the tablet’s pressure sensitivity feature affords a finer level of control over drawing dynamics [39]. This setup follows protocols similar to those used in previous studies, ensuring procedural consistency [2,29,38,39].

Each participant performed dot-to-dot tracing tasks of an equilateral triangle, measuring 150 mm per side on the monitor display, under 9 different randomized experimental conditions, which varied in the quantity of visual information provided, i.e., the number of points (visual cues) composing the displayed triangle. Specifically, the conditions were as follows: 3 (i.e., only the triangle vertices were presented), 6, 9, 12, 21, 30, 60, and 120 points, and full template (the complete template of the shape) [41]. The distances between the displayed points varied in each condition except for the full template: 150 mm, 75 mm, 50 mm, 37.50 mm, 21.43 mm, 15 mm, 7.50 mm, and 3.75 mm, respectively. For each condition, the shape rotated from 0 to 120 degrees with an interval of 15 degrees to prevent motor training or learning effects. Thus, the possible rotations were 0°, 15°, 30°, 45°, 60°, 75°, 90°, 105°, and 120°. The center of the triangle always coincided with the center of the monitor, and the rotations were fixed around the center of the shape. The number of points composing the shape and its rotation were presented in a randomized order, again to prevent motor training or learning effects. Participants were tested individually, completing one trial for each of the 9 conditions, with no specified rest time between conditions (Figure 1).

Before starting the experiments, the participant’s dominant hand was determined with the Edinburgh Handedness Inventory [42]. Then, participants were instructed to perform the experimental trial using their identified dominant hand. This precaution was taken because the triangle templates were designed to be used from medial to lateral with the dominant hand (i.e., counterclockwise for right-handed individuals) [29]. To familiarize themselves with the setup, before the start of the experiment, participants were instructed to draw random lines using the graphic pen tablet in the blank monitor window. During the trial, participants were seated without support for their wrist, arm, or elbow, ensuring that the only contact with the table was through the pen, consistent with the setup described by Cohen et al. (2019) [29]. Participants were briefed to look at the monitor and draw the triangle as precisely as possible, without focusing on the speed of execution [29]. According to Fitts’s Law, if a longer movement time is allowed, participants can afford to prioritize accuracy by making finer adjustments during the movement [43]. To guide participants in starting the dot-to-dot tracing task, numbers (‘1’, ‘2’, ‘3’) were positioned around each vertex of the triangle to indicate the order in which the sides should be traced (Figure 1). As the test lasted no longer than 10 min, participants did not experience cognitive or physical fatigue [25].

### 2.3. Analysis

The data were analyzed using an originally developed script in Python 3.9.

The absolute error was calculated by measuring the perpendicular distance from each data point on the drawn shape to the nearest point on the complete shape template and using the absolute value of this distance. This parameter reflects motor performance accuracy, with smaller values indicating lower levels of error. To implement population comparison, we compared the means of the absolute errors for each subject. Moreover, we computed the total time of execution, average speed, average smoothness, and average pressure for each shape drawn. The average speed was calculated as the distance covered along the drawn line divided by the total execution time. To assess average smoothness, acceleration was first derived from speed data, followed by the determination of jerk, which was derived from acceleration. The time-domain Savitzky–Golay filter [44] was applied to jerk data series to minimize noise. Finally, average smoothness was quantified by integrating the squared jerk values over time and normalizing them by the length of the speed data [45,46,47]. The average pressure was calculated as the mean of the tablet pressure data recorded continuously during each time interval of the line drawing. To identify outliers in our data, we conducted a boxplot analysis, leveraging distributional characteristics [48,49]. Also, the trendline for each parameter was calculated in order to better visualize the trend of data. Subsequently, Winsorization was performed by adjusting values at the 5th and 95th percentiles to mitigate the impact of outliers, matching them with predetermined percentiles of the dataset and promoting more robust and reliable analysis [49,50,51].

### 2.4. Statistics

Non-parametric analyses were conducted since the Shapiro–Wilk test was used a priori to determine the distribution of data, and they showed a non-normal distribution for our dataset (*p* > 0.05). To ensure that neither the degrees of rotation nor the order of execution affected tracing accuracy, the Friedman one-way repeated measures analysis of variance by ranks was implemented. This analysis compared absolute errors, times of execution, and pressures among conditions, with shape rotations and order of execution considered as dependent variables. Speed and smoothness were not included in this analysis, as they were parameters derived from time of execution. Furthermore, the Friedman test was also conducted separately for all the calculated parameters (absolute error, time of execution, speed, smoothness, and pressure) among experimental conditions (3, 6, 9, 12, 21, 30, 60, and 120 points, and full template). The Friedman test effect size was calculated using Kendall’s W coefficient of concordance (0.1–0.3 = small effect size; 0.3–0.69 = moderate effect size; 0.69–1.0 = large effect size). The Durbin–Conover post hoc test with Bonferroni correction was performed for pairwise comparisons of results with statistically significant differences. Spearman correlation was implemented for absolute error and time of execution, to explore the potential relationship between these variables (0.1–0.19 = negligible effect; 0.2–0.29 = weak effect; 0.3–0.39 = moderate effect; 0.4–0.69 = strong effect; 0.7–1 = very strong effect) [52,53].

## 3. Results

The percentage of Winsorized data for each parameter was as follows: 4.5% for the absolute error, 5.9% for time of execution, 2.7% for speed, 1.8% for smoothness, and 0% for pressure. These percentages were very low, indicating that the Winsorization had minimal impact on the overall data. Also, the Winsorized data did not differ substantially from the original data.

The Friedman test indicated that the rotation and order randomization had no significant effects on the absolute error [χ^2^(8) = 15.52, *p* = 0.11; χ^2^(8) = 16.03, *p* = 0.16, respectively], time of execution [χ^2^(8) = 2.45, *p* = 0.96; χ^2^(8) = 7.70, *p* = 0.46, respectively], and pressure [χ^2^(8) = 5.08, *p* = 0.75; χ^2^(8) = 37.45, *p* = 0.12, respectively].

The mean and standard deviation of absolute errors, times of execution, speed, smoothness, and pressure were reported across experimental conditions in Appendix A.

It was observed that, with the increase in visual information, the mean of absolute error decreased while the time of execution increased. Additionally, both speed and smoothness decreased with a higher amount of visual input. Pressure, however, remained constant across conditions. All these described results were statistically assessed using the Friedman test, as detailed further in Section 3.

Boxplots were presented to illustrate the comparative distribution of data for the absolute error (Figure 2A), for the time of execution (Figure 2B), for the speed (Figure 3A), for the smoothness (Figure 3B), and for the pressure (Figure 3C).

The distribution of the absolute errors decreases with increasing visual inputs (Figure 2A), consistent with the preliminary descriptive observations on means and standard deviations. The polynomial trendline reveals a non-linear reduction, characterized by a decline with a steeper slope initially, followed by a plateau beyond 21 points.

The distribution of the times of execution exhibits an opposing trend (Figure 2B), characterized by more linear increases in the polynomial trendline through conditions while evidently, the speed shows a mirror shape for the data distribution and trendline (Figure 3A).

The distribution of the smoothness through conditions (Figure 3B) has a decreasing trendline, although not strictly consistent.

The distribution of the pressure seems to remain constant across all experimental conditions (Figure 3C).

The comparison between the trendlines (Figure 2C) of absolute error (green) and the time of execution (orange) highlights an opposite trend of the two parameters and a reversal between 21 and 30 points, during which a convergence is observed. In Figure 2C, the standard error of the mean for both parameters is also presented as a shaded area around each trendline. The Spearman correlation results, indicated by the value above the trendline, show a moderate (0.3–0.39) to strong (0.4–0.69) correlation [52,53] between absolute error and time of execution across the condition from 9 points to full shape.

The Friedman test showed significant effects among experimental conditions in the absolute error [χ^2^(8) = 168.48, *p* < 0.001, W = 0.57], time of execution [χ^2^(8) = 92.21, *p* < 0.001, W = 0.31], speed [χ^2^(8) = 122.68, *p* < 0.001, W = 0.41], smoothness [χ^2^(8) = 96.49, *p* < 0.001, W = 0.39], with pressure being an exception [χ^2^(8) = 15.39, *p* = 0.052, W = 0.06]. Kendall’s W indicates strong concordance for absolute error and moderate concordance for time, speed, and smoothness.

The Durbin–Conover post hoc test showed a significant difference that first emerges from the subsequent conditions and then gradually stabilizes at varying levels, demonstrating a plateau effect starting from 12 points for absolute error (Figure 4A), time of execution (Figure 4B), and speed (Figure 4C). This plateau is indicated by an increasing number of non-significant comparisons (represented by yellow gradients). Beyond this point, the differences remain non-significant until the 120-point or full shape condition, thereby confirming the plateau effect. In contrast, differences in smoothness appear to be significant with at least a two-condition gap, except for 6- and 30-point conditions, demonstrating a less steep trend (Figure 4D).

## 4. Discussion

We investigated the impact of the amount of visual information on fine motor performance using a triangle dot-to-dot tracing task. Our results enlighten the relationship between the quantity of visual information and the inherent motor performance parameters of absolute error, time of execution, speed, smoothness, and pressure.

Absolute error was assessed as a measure of overall accuracy in task performance, consistent with the established literature [2,29,34,38]. We observed an inverse relationship between absolute error and the amount of visual information provided during the task. Specifically, as the quantity of visual cues increased (from 3 points to the full template), participants exhibited significantly reduced absolute error, indicating greater accuracy in reproducing the target shape. These results suggest that more extensive visual guidance enhances participants’ ability to accurately trace shapes. By emphasizing external cues, participants may effectively mitigate their reliance on internal cues, which have been identified as more susceptible to errors in fine motor tasks [2,32,33]. This phenomenon can be attributed to the inherent variability and subjectivity associated with internal cues, such as proprioceptive feedback and mental imagery [2]. Research indicates that individuals often struggle with maintaining accuracy when relying solely on these internal signals, leading to performance inconsistencies [2,32].

The reliance on visual feedback is particularly salient in tasks that require precision, such as tracing or drawing [2]. External cues provide a tangible reference point that can enhance motor control and execution accuracy [32,33]. As participants engage with visual feedback during these tasks, they can adjust their movements in real time, in line with error-driven learning, fostering a more adaptive learning environment [18,25].

This error-driven learning aligns with the concept of reinforcement learning, which involves modifying behavior based on the discrepancy between intended and actual outcomes (prediction errors) and underpins motor learning [18]. Visual feedback facilitates this process by allowing individuals to compare their movements with the target and make necessary adjustments. Our results suggest that increasing visual input does not necessarily lead to greater accuracy or adaptation. Rather, there appears to be a threshold beyond which additional visual feedback does not significantly enhance fine motor performance, as evidenced by the plateau observed in our data. This saturation of information helps prevent being overwhelmed by the volume of external information, thereby preventing a computational overload of the system.

Furthermore, our data show that the amount of visual input can be modulated, allowing individuals to refine their fine motor performance to varying degrees, depending on the level of guidance provided. This adaptability is crucial for mastering complex motor skills, as it enables real-time corrections based on external context, rather than relying solely on potentially flawed internal assessments. Therefore, this approach could serve as a non-invasive and ecological method to assess fine motor skills development in school-aged children, but it also offers an effective adaptive training strategy [5,13]. Additionally, healthy adults could use simple visual feedback exercises to improve their fine motor coordination, such as tracing a target on a screen, with the complexity of the target adjusted based on their performance. The level of visual guidance can be gradually reduced, encouraging participants to rely on internal motor control as they improve.

Externally cued movements preferentially involve the cerebellar and premotor circuits, while internally generated movements primarily engage the basal ganglia, pre-supplementary motor area (pre-SMA), and dorsolateral prefrontal cortex (DLPFC) [33]. Notably, individuals with Parkinson’s disease, characterized by basal ganglia dysfunction, show deficits in drawing tasks but often experience significant improvements in motor performance when external cues are introduced. This highlights the potential of tailored interventions that utilize visual feedback to enhance motor function by compensating for internal cueing deficits, for example, adaptive training that use virtual visual cues in tasks like writing or drawing. Struber et al. (2021) further support this by observing a reduction in absolute error during motor tasks, which correlated with EEG power variations in the beta band. These variations are linked to trial-to-trial error adaptations, suggesting that external cues may facilitate more efficient error correction processes. Moreover, their findings indicate that theta band variations are associated with learning processes, providing a comprehensive neural basis for the behavioral improvements observed when external guidance is introduced.

To further assess the impact of an increasing external guidance on motor performance, the time of execution was examined to reflect efficiency in performing the task and provide insight into cognitive processing speed during task completion [32,34,54]. Additionally, speed was derived from the time measurements to complement the analysis, focusing on how promptly movements were performed throughout the tracing task. We found a significant negative rank correlation between absolute error and time of execution across the experimental conditions (see Figure 2C). This implies that participants adopt a strategic approach to optimize performance by balancing speed and accuracy. As previously established, enhanced visual cues ultimately enabled participants to make real-time adjustments to their movements, although increasing the cognitive load of the process [9,38,39,43].

The present findings are in line with Woodworth’s [27] two-component seminal model and Fitts’s law [43], providing further evidence for a trade-off between movement accuracy (absolute error) and time of execution. Indeed, our participants could well prioritize accuracy over speed as they had no time limits to accomplish the task. Additionally, as the number of visual cues increased, the task complexity likewise increased [32,33,43], leading participants to require more time to complete the tracing. This reduction in drawing speed also aligns with Fitts’s law, which predicts that task complexity is positively related to the time needed to perform an action [43]. This relationship is evident in studies where children’s performance on pointing tasks varies with task complexity, highlighting the importance of adapting learning environments to match developmental stages [2,55]. For instance, younger children may benefit from simpler tasks before gradually introducing more complex elements [55], and the motor performance evaluation can be supported by analyzing the time of execution to assess progress.

Similarly to the absolute error, a plateau is also observed in execution time starting from the 12-point condition. This suggests that there is a threshold beyond which additional visual feedback does not significantly increase the time required to perform the task. This saturation appears to prevent an excessive slowdown of the system, which could otherwise become dysfunctional. The maximum time required to perform a task with the maximum accuracy could be an extremely important parameter to analyze the fine motor performance. For example, a tracing task involving full shape figures, where there are no time limits, could be used to evaluate performance. In a training context, healthy adults could practice this task with gradually increasing complexity, such as tracing figures with more intricate shapes or smaller targets, while maintaining accuracy. This approach allows for the assessment of fine motor coordination and the development of more efficient movement patterns.

Alongside the traditional measurements, we also analyzed smoothness, which assesses the fluency of movements. Smoothness is a key indicator of motor coordination and control, with higher scores suggesting better integration of cognitive processes and motor execution. Previous studies have shown that smoothness is a critical parameter for assessing motor skill proficiency [32,34,46,56]. This analysis provides insight into the quality of movement, which is crucial for understanding fine motor performance.

The smoothness of performance decreased as the amount of visual information increased, indicating that participants execute movements with greater fluency and coordination when fewer visual cues were present. This outcome aligns with previous findings that the frequency of movement adjustments increases with task complexity due to increased visual input [32,34,56]. It has been suggested that the intermittence of movement, caused by the increasing number of visual cues, may reflect the escalating challenges in eye–hand coordination [32,34,45,46,56]. We found that the smoothness reached a plateau beyond a certain threshold from around 12 points, favoring the hypothesis of a cut-off rate in the frequencies of motion adjustments (see Figure 3B). However, this effect was less clear in the 6-point condition. Notably, this plateau threshold for smoothness is less pronounced compared to the plateau observed in both absolute error and execution time.

Overall, our results emphasize the importance of smoothness as a critical indicator of motor skill proficiency, revealing how cognitive processes interact with motor execution. This highlights the potential of using smoothness for evaluating motor proficiency in children, particularly in developmental assessments where smoother movements reflect more advanced motor control and better coordination [45,46].

Furthermore, our results align with the findings of Yabuki et al. (2020), which discuss how error correction during movement impacts smoothness. It is emphasized that qualitative assessments of movement, such as smoothness, provide insights that quantitative measures alone cannot capture [57]. Similarly, healthy adults could practice handwriting or tracing tasks with varying complexity to promote smoother and more efficient movements as their motor skills develop. Future research should consider our findings when evaluating this parameter in the context of motor learning assessment [16,45].

Pressure applied during tracing was evaluated as a proxy for grip strength and control, which are important factors in tailoring rehabilitation strategies for individuals with fine motor impairments [37,39,58]. The distribution of pressure applied during tracing remained consistent across experimental conditions, suggesting that participants maintained a steady level of pen pressure, regardless of the amount of visual information provided. This non-significant result indicates that, unlike other parameters such as absolute error or time of execution, pressure may be less sensitive to variations in task complexity or visual input. This consistency may reflect a baseline motor strategy in which grip control remains stable, not requiring the same level of adaptation as other performance aspects (e.g., accuracy or speed). These findings are consistent with previous research showing that neither pen-tip normal force nor total grip force signals are significantly influenced by changes in task conditions [37,39]. For instance, Gatouillat et al. (2017) found that neither speed nor timing constraints impacted pen-tip normal force or total grip force signals [39]. However, other writing features were influenced by speed and timing constraints, confirming the trade-offs between speed–accuracy and timing–accuracy [39]. This strategy allows individuals to maintain a reliable grip on the pen, crucial for executing fine motor tasks accurately. By stabilizing grip pressure, participants may minimize the risk of losing control over the writing instrument, ensuring a baseline level of performance regardless of task complexity. Another possibility is that participants rely on proprioceptive and tactile feedback to regulate their grip pressure. Future studies could explore whether pen pressure remains constant during fine motor tasks in both pathological populations and typically or atypically developing children, and under different conditions.

One limitation of our study is that it was focused on a specific motor task (triangle dot-to-dot tracing) with predetermined shapes and points. As a consequence, our findings may not fully cover the paramount complexities of real-world fine multifarious motor activities, which often involve varying degrees of accuracy and coordination in predictable and unpredictable settings. Future studies are therefore encouraged to analyze a broader range of shapes and incorporate additional input characteristics to reflect a more complete picture of real-world scenarios.

Moreover, due to the novelty of the testing procedure, we acknowledge that the test–retest reliability was not assessed, and we recommend that future research investigate this aspect to ensure the stability and consistency of the test over time.

Also, it is important to consider that this study lacks an analysis of anthropometric profiles and sex differences, which are known to influence motor and postural control [8,23,34]. Future research should consider these variables using more diverse samples to better understand their impact on visual biofeedback effects.

While accuracy is already well-defined in the context of fine motor activities [29,34,38,39], we also investigated other motor performance parameters such as speed, smoothness, and pressure. Thus, future studies are stimulated to explore additional motor performance parameters to gain a more comprehensive understanding and a multifaceted dimension of fine motor behavior.

## 5. Conclusions

In conclusion, this study advances our understanding of how sensory cues influence fine motor performance. Our results showed that increasing visual input improved accuracy (reduced absolute error) but increased execution time and decreased smoothness, highlighting a trade-off between speed and accuracy as task complexity increased. Our outcomes evidence a threshold beyond which additional visual feedback does not significantly enhance fine motor performance. The saturation of external information plays a crucial role in optimizing motor performance, ensuring that the system operates efficiently without being overwhelmed, thus enhancing motor control and cognitive processing. These findings could inform protocols for fine motor learning to enhance coordination and dexterity [13]. Furthermore, they may support the development of effective rehabilitation strategies, particularly for individuals with motor coordination deficits or neurological impairments [19,59].

The consistent application of pressure during tracing suggests that grip control remains stable across varying task complexities, highlighting the importance of maintaining a reliable motor strategy for fine motor tasks and offering valuable insights for rehabilitation approaches targeting grip strength and control.

By examining the multidimensional interplay between sensory visual information and motor behavior, this study contributes to optimizing training protocols and rehabilitation interventions based on a more precise understanding of sensorimotor integration.

## Figures and Tables

**Figure 1 jfmk-09-00267-f001:**
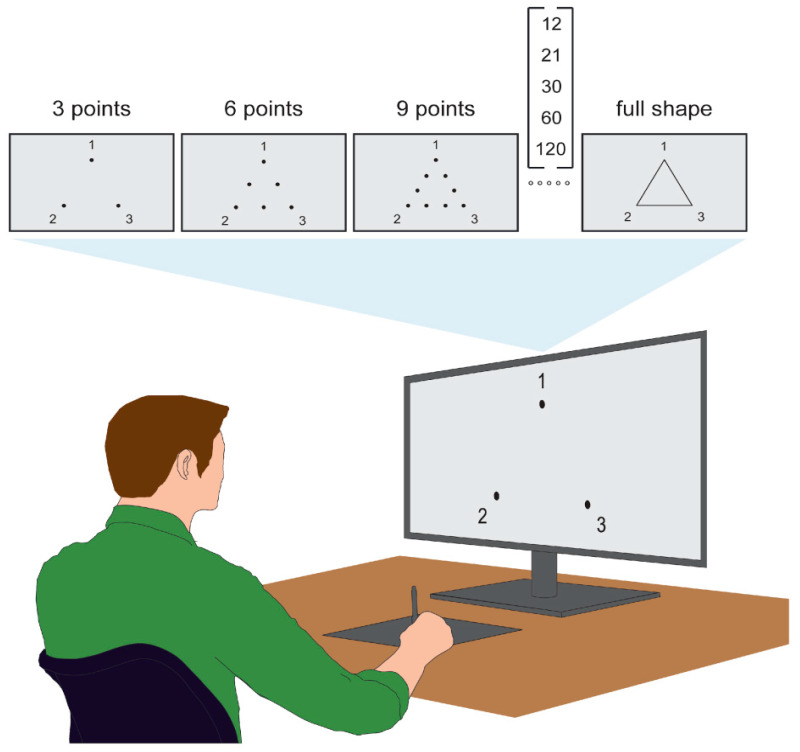
Experimental setup. The upper gray rectangles represent the conditions displayed to the participant. The conditions not represented between 9 points and full shape are summarized inside square brackets. Numbers ‘1’, ‘2’, ‘3’ were positioned around each vertex of the triangle to indicate the order in which the sides should be traced. Other informations are in the text. Below is the experimental setup. In the figure, a stylized depiction shows the seated subject drawing on a tablet while looking at the monitor.

**Figure 2 jfmk-09-00267-f002:**
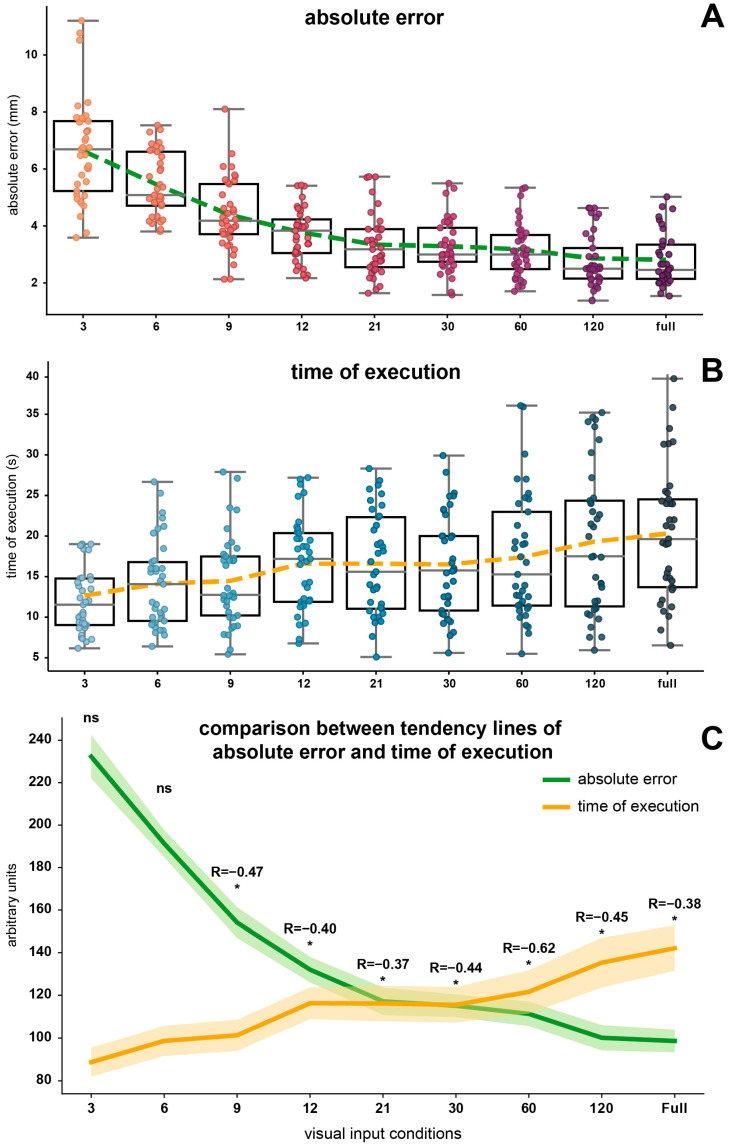
Boxplots representing the distribution of absolute error (**A**) and time of execution (**B**) across the experimental conditions. The dotted lines indicate the trendlines, obtained by connecting the means of each condition. (**C**) Comparison between the trendlines of absolute error (green) and time of execution (orange). The shaded areas around both trendlines represent the standard error of the mean. The values above the trendlines indicate the Spearman correlation coefficient R. The ‘*’ indicates that the Spearman correlation test is statistically significant, while ‘ns’ indicates no significant correlation.

**Figure 3 jfmk-09-00267-f003:**
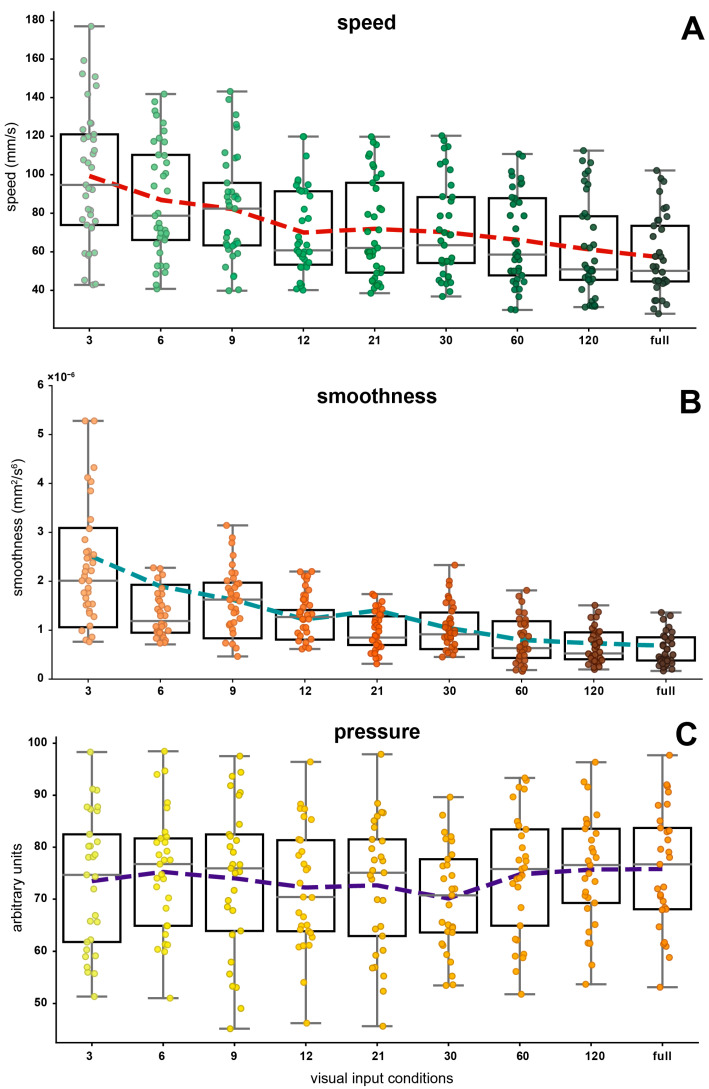
Boxplots representing the distribution of speed (**A**), smoothness (**B**), and pressure (**C**) across the experimental conditions. Each color gradation of the circles inside the boxplots corresponds to a different experimental condition. The dotted lines indicate the trendlines, obtained by connecting the means of each condition.

**Figure 4 jfmk-09-00267-f004:**
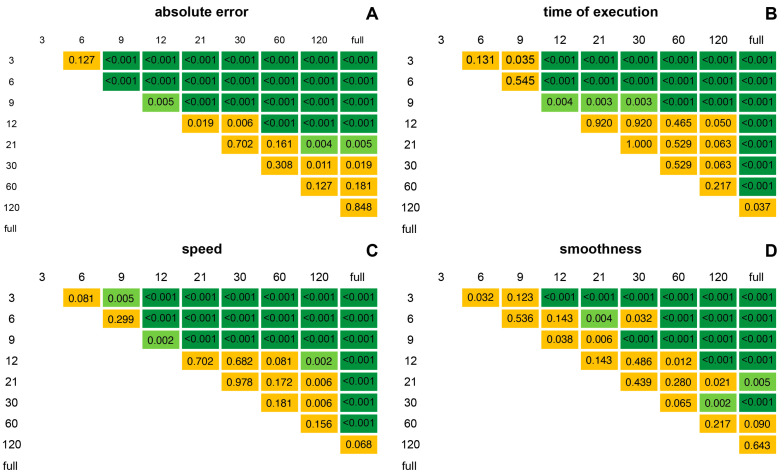
Triangular heatmap of pairwise comparisons using Durbin–Conover post hoc analysis with Bonferroni correction (*p*-values = 0.0056) of absolute error (**A**), time of execution (**B**), speed (**C**), and smoothness (**D**). Dark green indicates *p*-values < 0.001, light green indicates *p*-values between 0.001 and 0.0056, and yellow indicates non-significant *p*-values > 0.0056.

## Data Availability

Dataset available on request from the authors.

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
