# Peer review of "The Role of Visual Information Quantity in Fine Motor Performance"

_jfmk, 2024, doi:10.3390/jfmk9040267_

Round 1
Reviewer 1 Report
Comments and Suggestions for Authors
Dear authors, the document entitled “The Role of Visual Information Quantity in Fine Motor Performance” is very interesting and nicely written. Some suggestions are made to improve the overall quality of the document,
In the phrase “It was observed that, with the increase in visual information, the mean of absolute error decreased while the time of execution increased. Additionally, both speed and smoothness decreased with a higher amount of visual input. Pressure, however, remained constant across conditions.” – the statistical tests to corroborate these claims should be embedded in the same paragraph.
It seems to me that the information between Table 1 and Figure 2 is the same, if so, why replicate the same information in a different presentation?
At last, I would suggest that the conclusions section includes a summary of your findings and directions for future studies.
Kind regards
Author Response
Dear authors, the document entitled “The Role of Visual Information Quantity in Fine Motor Performance” is very interesting and nicely written. Some suggestions are made to improve the overall quality of the document,
Comment 1: In the phrase “It was observed that, with the increase in visual information, the mean of absolute error decreased while the time of execution increased. Additionally, both speed and smoothness decreased with a higher amount of visual input. Pressure, however, remained constant across conditions.” – the statistical tests to corroborate these claims should be embedded in the same paragraph.
Response 1: Thank you for your comment. To address your concern, we have modified the text to include a direct mention of the statistical analysis in line 226-230.
Comment 2: It seems to me that the information between Table 1 and Figure 2 is the same, if so, why replicate the same information in a different presentation?
Response 2: Thank you for pointing out the potential redundancy between Table 1 and Figure 2. We have decided to delete Table 1 from the manuscript to avoid unnecessary redundancies of information. However, we have added Table 1 as supplementary material, as it contains explicit numeric data that can help the reader. We appreciate your feedback, which helped enhance the quality of our work.
Comment 3: At last, I would suggest that the conclusions section includes a summary of your findings and directions for future studies.
Response 3: Thank you for your valuable suggestion. We have revised the conclusion section to include a summary of our key findings, while the future studies were indicated at the end of discussion section.
Reviewer 2 Report
Comments and Suggestions for Authors
Dear Authors,
First of all, congratulations to the authors for conducting such interesting and apparently well-developed research. Moreover, it has a moderate clinical/practical impact.
However, the manuscript has formal errors and methodological limitations that should be addressed before possible publication in this Journal.
ABSTRACT:
Concrete QUANTITATIVE results should be included.
INTRODUCTION:
The Authors should include fundamental references about their object of study that they have omitted (DOI: 10.3390/diagnostics11040637).
METHODS:
The research design should be stated at the beginning of the section.
RESULTS:
The statistical power of the final sample analysed should be stated.
DISCUSSION:
This section would also benefit from the inclusion of reference works such as those recommended for the Introduction.
Kind regards
Author Response
Dear Authors,
First of all, congratulations to the authors for conducting such interesting and apparently well-developed research. Moreover, it has a moderate clinical/practical impact. However, the manuscript has formal errors and methodological limitations that should be addressed before possible publication in this Journal.
Comment 1:
ABSTRACT:
Concrete QUANTITATIVE results should be included.
Response 1: Thank you for your suggestion. We have now included quantitative results in the abstract as requested (lines 17-21).
Comment 2:
INTRODUCTION:
The Authors should include fundamental references about their object of study that they have omitted (DOI: 10.3390/diagnostics11040637).
Response 2:
We would like to thank you for your suggestion. We have reviewed the manuscript and included this key reference in both introduction (line 49) and discussion (line 447).
Comment 3:
METHODS:
The research design should be stated at the beginning of the section.
Response 3:
Thank you for your comment. We have revised the Methods section to address your comment. The research design is now clearly stated at the beginning of the section to provide a more coherent structure (lines 109-110).
Comment 4:
RESULTS:
The statistical power of the final sample analysed should be stated.
Response 4:
Thank you for your comment. We have included the statistical power for the final sample size (line 114).
Comment 5:
DISCUSSION:
This section would also benefit from the inclusion of reference works such as those recommended for the Introduction.
Response 5:
We would like to sincerely thank you for your comment. We have reviewed the discussion paragraph and included the key reference you suggested (line 447), as well as others.
Reviewer 3 Report
Comments and Suggestions for Authors
Introduction
I think that the authors should further explain and strengthen the rationale for selecting triangle tracing as the task. Why not selecting real-world tasks, for example handwriting? Please further explain it.
The authors should further develop the overall introduction section by focusing more on the study specific contribution instead of the broad importance of fine motor skills. Please improve the section.
Please provide the hypothesis for the study aim.
Please provide more info and details on the “sensorimotor integration” definition and the relevance for the study.
Methods
The authors state that “This analysis indicated that a minimum of 34 participants was necessary to detect a medium effect size with 0.80 power and a 0.05 significance level, justifying the inclusion of 37 participants in this study”. What did the authors mean with “medium effect size”? Based on which comparison? Is it based on previous studies or a pilot intervention? Please you should strengthen the choice made.
Why only right-handed participants? Please provide a rationale.
The authors should further strengthen and justify the potential confounding factors like hand dominance, familiarity with digital tools, or fatigue due to continuous trials. Please provide the rationale or evidence regarding the choice made.
Is there any reliability and validity of the testing procedures? Please provide some evidence on the reliability, validity and accuracy of the testing procedures and device adopted.
Results
How much of the data was Winsorized? Please provide absolute values as well not only the percentiles. How did this affect the results? Please provide more info.
I think that the authors should perform and report a statistical cut-off, sensitivity and specificity analysis to validate the plateau effect in absolute error and other metrics. Please provide more details and data analysis to further strengthen your results.
In line with other researches on the effect of visual biofeedback on motor and postural control, with variations observed across different anthropometric profiles and sexes, the authors should probably further check their analysis and control these variables. Please improve it.
Discussion
The authors should further highlight the non-significant results for pressure. Please provide more explanations and assumptions. This might be an important finding.
The authors should make an effort to further elaborate new theories.
Furthermore, the discussion of “adaptive learning” is quiet vague and inconsistent and does not connect back to the specific results. Please improve it.
Practical examples of training and exercises should be provided. Concrete examples are needed. This might be needed also for other populations mentioned, although it is speculative. Probably the authors should mainly focus on the practical applications within the specific population assessed and afterwards suggesting some practical applications for different populations that have not been assessed.
Limitations and future research should also be proposed and acknowledged. Please expand for the readers.
Finally, the manuscript will definitely benefit from a more rigorous explanation of the plateau effect and its implications for motor control theories, training, exercises and testing within the assessed population.
Author Response
Comment 1:
Introduction
I think that the authors should further explain and strengthen the rationale for selecting triangle tracing as the task. Why not selecting real-world tasks, for example handwriting? Please further explain it.
Response 1:
Thank you for your comment. We appreciate your suggestion to clarify the rationale for selecting the triangle tracing task. We have provided a more detailed explanation from lines 67 to 81. Here is the updated explanation:
“Among the various tracing tasks used in the literature, we selected the triangle tracing task for several key reasons aligned with the aim of our study. First, the triangle represents the simplest 2D geometric form, offering a controlled and simplified method to study the influence of varying levels of visual feedback on motor performance. Its simplicity allows for more precise measurements of movement accuracy and control, without the confounding factors present in more complex, real-world tasks like handwriting. Second, triangle tracing tasks are highly reproducible and suitable for experimental manipulation, such as varying the amount of visual input and shape rotation, both of which are central to our study. Unlike handwriting, which can be influenced by individual writing styles and cognitive factors like language or memory, triangle tracing offers more consistent motor performance across participants. This consistency is crucial for isolating the effects of visual feedback on motor behavior. Additionally, previous studies have successfully used triangle tracing tasks to explore fine motor performance, making it a well-established model for investigating the effects of visual feedback on motor control [20, 21, 46]”
We hope this explanation clarifies our choice of task and the rationale behind it. We also appreciate your feedback, which has helped improve the quality of our work.
Comment 2:
The authors should further develop the overall introduction section by focusing more on the study specific contribution instead of the broad importance of fine motor skills. Please improve the section.
Please provide the hypothesis for the study aim.
Please provide more info and details on the “sensorimotor integration” definition and the relevance for the study.
Response 2:
Thank you for your valuable suggestions. We have revised the introduction to address all the advices. Specifically, we have highlighted the specific contributions of this study (lines 100-106), explicitly included the hypothesis for the study's aim (lines 94-99), and provided a detailed definition of sensorimotor integration and its relevance (lines 57-60).
Comment 3:
Methods
The authors state that “This analysis indicated that a minimum of 34 participants was necessary to detect a medium effect size with 0.80 power and a 0.05 significance level, justifying the inclusion of 37 participants in this study”. What did the authors mean with “medium effect size”? Based on which comparison? Is it based on previous studies or a pilot intervention? Please you should strengthen the choice made.
Response 3:
Thank you for your valuable comment. We would like to clarify and correct an error: a small effect size was considered, not a medium one. The term small effect size refers to the standardized effect size defined by Cohen (1988), where a small effect size corresponds to Cohen’s f = 0.2 for ANOVA-type tests. This effect size was chosen based on the within-subject experimental design of the study, which involved 9 repetitions per participant, and the model was planned to be a repeated measures ANOVA with within-subject factors. The value was used during the power analysis conducted in G*Power 3.1 software to determine the minimum sample size required to achieve sufficient statistical power (0.80) at a significance level of 0.05. Upon reviewing the results, we found that the minimum sample size required was actually 22, so we have corrected this as well. In the absence of directly comparable prior studies for this specific experimental design, we adopted a conservative approach by assuming a small effect size to ensure an adequate sample size for detecting meaningful differences across conditions. Additionally, we considered sample sizes reported in the literature for behavioral and motor performance studies involving tasks requiring precision and coordination (Shafer et al., 2019; Vaillancourt et al., 2002; Yabuki et al., 2020; Yamamoto et al., 2022). We have added further explanation to the text (lines 113 and 114-116).
Comment 4:
Why only right-handed participants? Please provide a rationale.
Response 4:
Thank you for your insightful comment. The study focused on right-handed participants because the triangle templates were specifically designed for a medial-to-lateral movement using the dominant hand (i.e., counterclockwise for right-handed individuals). Including left-handed participants could have introduced additional variability in the data due to differences in motor skills, spatial reasoning, and cognitive processing related to handedness. By restricting the sample to right-handed participants, we aimed to reduce this potential confounding factor and ensure more consistent performance across the group. We have added further clarifications in lines 160-162.
Comment 5:
The authors should further strengthen and justify the potential confounding factors like hand dominance, familiarity with digital tools, or fatigue due to continuous trials. Please provide the rationale or evidence regarding the choice made.
Response 5:
Thank you for your comment. We asked the participants if they had experience with this type of graphic pen, and all of them indicated that they were naïve to the setup. We have added this information in lines 118-119.
As the test lasted no longer than 10 minutes, participants did not experience cognitive or physical fatigue (doi: 10.1093/cercor/bhae085). We have added this information in lines 173-174.
Comment 6:
Is there any reliability and validity of the testing procedures? Please provide some evidence on the reliability, validity and accuracy of the testing procedures and device adopted.
Response 6:
Thank you for your insightful comment. The testing devices used in this study were carefully selected and validated to ensure reliability and accuracy. The graphic pen tablet (Wacom Intuos® CTH-690AK) used for the tracing task allows for precise and consistent measurements. The use of a high-frequency pen tablet (133 Hz) and a high-resolution 1080p monitor ensures precise capture of pen movements. Furthermore, the tablet's 1024 pressure sensitivity levels and 2540 lpi resolution provide consistent accuracy in measuring pressure parameters across trials. This setup follows similar protocols used in previous studies (e.g., Cohen et al., 2018A, Cohen et al., 2018B, Cohen et al., 2019, Gatouillat et al., 2017), ensuring procedural consistency (see lines 134-135).
Regarding the test, due to the novelty of the procedure, we acknowledge in the limitations section the lack of test-retest reliability, which should be addressed in future studies (see lines 443-445). However, the task design, including the number of trials, randomized conditions, and rotations, mitigates learning effects, contributing to the reliability of the results.
Comment 7:
Results
How much of the data was Winsorized? Please provide absolute values as well not only the percentiles. How did this affect the results? Please provide more info.
Response 7:
The Winsorized data were respectively: 4.5 % for the absolute error, 5.9 % for the time of execution, 2.7 % for speed, 1.8 % for the smoothness, and 0% for the pressure. These percentages were very low. Also, the winsorized data were not substantially different from the results obtained without Winsorization at the upper and lower α thresholds, indicating that the adjustments did not significantly affect the outcomes. We have added this information in lines 217-221.
Here, we also provide the absolute values of the 95th and 5th percentiles. Since these values are already represented by the whiskers of the box plot, we chose not to include them in the main text to avoid redundancy.
|
Absolute error |
3 |
6 |
9 |
12 |
21 |
30 |
60 |
120 |
full |
|
Min Value |
3.59 |
3.81 |
2.13 |
2.17 |
1.64 |
1.58 |
1.71 |
1.38 |
1.54 |
|
Max Value |
11.20 |
7.53 |
8.10 |
5.75 |
6.15 |
5.50 |
5.96 |
5.46 |
5.02 |
|
Time of execution |
3 |
6 |
9 |
12 |
21 |
30 |
60 |
120 |
full |
|
Min Value |
6.17 |
6.40 |
5.43 |
6.77 |
5.10 |
5.60 |
5.50 |
5.93 |
6.53 |
|
Max Value |
25.83 |
26.68 |
27.89 |
27.19 |
28.31 |
29.92 |
36.07 |
35.21 |
39.80 |
|
speed |
3 |
6 |
9 |
12 |
21 |
30 |
60 |
120 |
full |
|
Min Value |
41.14 |
40.78 |
39.81 |
40.13 |
38.53 |
36.83 |
29.86 |
31.29 |
27.87 |
|
Max Value |
176.99 |
141.84 |
143.18 |
119.78 |
119.70 |
120.21 |
110.76 |
112.51 |
102.18 |
|
Pressure |
3 |
6 |
9 |
12 |
21 |
30 |
60 |
120 |
full |
|
Min Value |
28.01 |
31.35 |
25.55 |
29.80 |
26.81 |
37.82 |
29.29 |
36.97 |
34.14 |
|
Max Value |
98.31 |
98.46 |
97.53 |
96.42 |
97.88 |
89.64 |
93.33 |
96.36 |
97.70 |
|
Smoothness |
3 |
6 |
9 |
12 |
21 |
30 |
60 |
120 |
full |
|
Min Value |
9.90E-07 |
8.09E-07 |
4.63E-07 |
6.15E-07 |
3.13E-07 |
4.52E-07 |
1.85E-07 |
1.97E-07 |
1.67E-07 |
|
Max Value |
5.27E-06 |
2.21E-06 |
3.11E-06 |
2.14E-06 |
1.88E-06 |
2.24E-06 |
1.92E-06 |
1.53E-06 |
1.39E-06 |
Comment 8:
I think that the authors should perform and report a statistical cut-off, sensitivity and specificity analysis to validate the plateau effect in absolute error and other metrics. Please provide more details and data analysis to further strengthen your results.
Response 8:
Thank you for your constructive critique. We have enhanced the results section to provide a more detailed explanation of the plateau effect, particularly as illustrated in Figure 4. Specifically, we used the heatmap of pairwise comparisons from the Durbin-Conover post-hoc analysis to validate the statistical cut-off. The analysis revealed that for absolute error, time of execution, speed and smoothness, a plateau begins to emerge starting at the 12-point condition. This is evidenced by the increasing number of non-significant comparisons (highlighted by yellow gradients on the heatmap). Beyond the 12-point condition, the differences remain non-significant until the 120-point or full shape condition. This pattern indicates a significant reduction in the steepness of the performance curve, thereby confirming the plateau effect.
We have clarified this point in the manuscript (lines 272–276) to strengthen our results and align with your suggestion.
We appreciate your feedback, which has helped improve the quality of our work
Comment 9:
In line with other researches on the effect of visual biofeedback on motor and postural control, with variations observed across different anthropometric profiles and sexes, the authors should probably further check their analysis and control these variables. Please improve it.
Response 9:
Thank you for your valuable comment. We agree that anthropometric profiles and sex differences can significantly influence motor and postural control. While these variables were not the primary focus of our study, we acknowledge their potential impact on the results. To address this, we have included a discussion of this limitation in the manuscript (lines 446-449) and emphasized the need for future research to include larger and more diverse samples to better account for these factors.
Comment 10:
Discussion
The authors should further highlight the non-significant results for pressure. Please provide more explanations and assumptions. This might be an important finding.
Response 10:
We sincerely thank you for your valuable feedback. In the revised manuscript, we have significantly expanded the discussion of the non-significant results for pressure between lines 418 and 435.
Comment 11:
The authors should make an effort to further elaborate new theories.
Furthermore, the discussion of “adaptive learning” is quiet vague and inconsistent and does not connect back to the specific results. Please improve it.
Practical examples of training and exercises should be provided. Concrete examples are needed. This might be needed also for other populations mentioned, although it is speculative. Probably the authors should mainly focus on the practical applications within the specific population assessed and afterwards suggesting some practical applications for different populations that have not been assessed.
Response 11:
Thank you for your constructive feedback. We appreciate your suggestion to further elaborate on new theories, and we have made efforts to clarify and expand on the theoretical framework in the revised manuscript. Regarding the discussion of 'adaptive learning,' we acknowledge that it was somewhat vague in the initial draft. We have revised this section to make the connection between adaptive learning and the specific results much clearer. Additionally, we expanded on how adaptive learning relates to the findings, providing a more consistent and coherent interpretation of the data. We have improved the entire discussion section by adding new theories and practical examples, particularly focused on healthy subjects (see lines 308-327, 370-381, 408-410).
We are grateful for your valuable contribution, which has undoubtedly strengthened our manuscript.
Comment 12:
Limitations and future research should also be proposed and acknowledged. Please expand for the readers.
Response 12:
Thank you for your comment. We have expanded the parts of limitation and future studies (lines 443-449).
Comment 13:
Finally, the manuscript will definitely benefit from a more rigorous explanation of the plateau effect and its implications for motor control theories, training, exercises and testing within the assessed population.
Response 13:
We would like to sincerely thank you for your observation. We agree that a more detailed explanation of the plateau effect would improve the manuscript. In the revised version, we have expanded on the plateau effect and try to extrapolate new theories. We believe these revisions enhance the manuscript and address your concern.
Thank you once again for your suggestion to improving the quality of our manuscript.